# Childhood Asthma Biomarkers Derived from Plasma and Saliva Exosomal miRNAs

**DOI:** 10.3390/ijms26157043

**Published:** 2025-07-22

**Authors:** Abdelnaby Khalyfa, Mohit Verma, Meghan M. Alexander, Zhuanhong Qiao, Tammy Rood, Ragini Kapoor, Trupti Joshi, David Gozal, Benjamin D. Francisco

**Affiliations:** 1Department of Biomedical Sciences, Joan C. Edwards School of Medicine, Marshall University, Huntington, WV 25701, USA; mvermam@marshall.edu (M.V.); joshitr@marshall.edu (T.J.); 2Marshall Institute for Interdisciplinary Research, Marshall University, Huntington, WV 25703, USA; 3Department of Pediatrics, School of Medicine, University of Missouri, Columbia, MO 65211, USA; alexandermm@missouri.edu (M.M.A.); qiaoz@health.missouri.edu (Z.Q.); roodtl@missouri.edu (T.R.); kpoorra@health.missouri.edu (R.K.); 4Departments of Pediatrics and Biological Sciences, Joan C. Edwards School of Medicine, Marshall University, Huntington, WV 25755, USA; gozal@marshall.edu

**Keywords:** asthma, exosomes, extracellular vesicles, plasma, saliva, miRNAs, lncRNAs

## Abstract

Asthma, the most common chronic respiratory condition in children, involves airway inflammation, hyper-responsiveness, and frequent exacerbation that worsen the airflow and inflammation. Exosomes, extracellular vesicles carrying microRNAs (miRNAs), play a key role in cell communication alongside other types of communication and are promising markers of asthma severity. This study compares exosomal miRNA and long non-coding RNA (lncRNA) profiles in boys with asthma, focusing on differences between those with normal lung functions and those with severe airflow obstruction. This study enrolled 20 boys aged 9–18 years with asthma, split into two groups based on their lung function. Ten had normal lung function (NLF; FEV1/FVC > 0.84, FEF75% > 69% predicted), while ten had severe airflow obstruction (SAO; FEV1/FVC < 0.70, FEF75 < 50% predicted). Saliva and blood samples were collected. Exosomes were isolated, quantified, and analyzed via small RNA sequencing to identify differentially expressed (DE) miRNA and lncRNA profiles. Bioinformatic tools were then used to explore potential miRNA biomarkers linked to asthma severity. SAO subjects were more likely to exhibit allergen sensitization, higher IgE levels, and more eosinophils. We identified 27 DE miRNAs in plasma and 40 DE miRNAs in saliva. Additionally, five key miRNAs were identified in both saliva and plasma which underline important pathways such as neurotrophins, T-cell receptor, and B-cell receptor signaling. We further outlined key features and functions of miRNAs and long non-coding RNAS (lncRNAs) and their interactions in children with asthma. This study identified DE miRNAs and lncRNAs in children with SAO when compared to those with NLF. Exosomal miRNAs show strong potential as non-invasive biomarkers for personalized asthma diagnosis, treatment, and monitoring. These RNA markers may also aid in tracking disease progression and response to therapy, thereby supporting the need for future studies aimed at applications in precision medicine.

## 1. Introduction

Asthma, the most common chronic pediatric respiratory disease, is characterized by chronic eosinophilic airway inflammation, variable airflow limitations, and symptoms such as wheezing, shortness of breath, chest tightness, and coughing [1,2]. Asthma prevalence revolves around 8.3% of children in the United States [3], and affects over 300 million people globally, with U.S.A’s costs exceeding USD 85 billion annually. Epidemiological studies have linked elevated asthma prevalence to several major risk factors, including obesity, allergic rhinitis, atopy, a minority race or ethnicity, and one’s socioeconomic status [4]. Spirometry, including the post-bronchodilator response, is key for the diagnosis and helps to distinguish airway obstruction from other respiratory conditions [2]. Appropriate asthma management has a major impact both on the quality of life of patients and their families, as well as on public health outcomes [5]. Genetic and environmental factors influence the prevalence and severity [6], but efforts to predict the long-term asthma risk through phenotyping and genetic markers have so far shown limited clinical use. Pediatric asthma is defined by chronic airway inflammation, hyper-responsiveness, and intermittent—sometimes persistent—airflow obstruction [7]. Despite available treatments, many children continue to experience uncontrolled symptoms and frequent exacerbations, contributing to long-term deterioration in lung function accompanied by increased healthcare utilization.

This complex, heterogeneous disease involves the imbalance of both innate and adaptive immune cells that interact with airway epithelial cells, leading to bronchial hyper-reactivity and airway remodeling in response to allergens, infections, or pollutants [8]. Its complexity is further enhanced by the presence of various phenotypes (i.e., a clinical presentation, patterns of progression, and treatment responses) as well as by various endotypes (divergent mechanisms) [9,10]. The most common asthma variant in children, T2-high asthma, is characterized by atopy, eosinophilic inflammation, elevated Th2 cytokines (IL-4, IL-5, and IL-13), and leukocyte recruitment driven by CD4 + T-cells, mast cells, and eosinophils [9,10,11]. Allergen exposure triggers immunoglobulin E (IgE)-mediated early-phase bronchoconstriction, often followed by a late-phase response [12]. Activated mast cells release histamine and leukotrienes, while IL-5 drives eosinophil recruitment. Chronic inflammation causes repeated tissue injury and airway remodeling through complex immune signaling [13]. In children with severe asthma, T2 inflammation is common and associated with exacerbations, impaired lung function, and a reduced quality of life, even though the advent of biologic therapies has led to significantly improved outcomes [14,15,16].

Current tools for diagnosing and monitoring asthma include spirometry, symptom questionnaires, and the measurement of circulating inflammatory markers such as blood eosinophil counts and serum IgE levels. Activated mast cells release histamine and leukotrienes, and IL-5 recruits eosinophils; together, they drive chronic inflammation that causes tissue damage and airway remodeling through immune signaling [17,18]. Particularly in pediatric populations, these markers are often nonspecific and fail to distinguish between overlapping clinical phenotypes. There is a clear need for non-invasive, reliable biomarkers that reflect disease severity, point to underlying immune mechanisms, and guide treatment, especially in children where standard testing may be limited or inconsistent [19].

Exosomes are small vesicles (30–150 nm) that transfer miRNAs, mRNAs, and proteins between cells, influencing function, and are a promising source of biomarkers [20]. Exosomes found in nearly all biological fluids—including blood, urine, saliva, breast milk, cerebrospinal fluid, and amniotic fluid—are involved in maintaining cell homeostasis and reflect the molecular profile of their parent cells [21,22,23]. Exosomes’ role in modulating recipient cell behavior links them to the pathogenesis of inflammatory and respiratory diseases [21,23,24]. Due to these properties, exosomes are being explored as promising tools for liquid biopsy and therapeutic applications [25,26]. Exosomal miRNAs play a key role in asthma by regulating inflammation, airway hyper-responsiveness, and remodeling. Since each miRNA can target hundreds of genes, changes in miRNA levels can disrupt multiple pathways and significantly impact disease development [27]. Consequently, alterations in miRNA expression could participate in the initiation and progress of asthma, as well as its clinical expression [28]. Furthermore, distinct miRNA expression profiles have been observed between asthma phenotypes and in relation to the clinical severity, supporting their potential role in endotyping and risk stratification [29]. However, much of this work has focused on adult populations or bulk tissue analysis. Thus, our study aimed to investigate exosomal miRNA and lncRNA expression differences in two groups of children with vastly different clinical phenotypes.

To this effect, the present study focused on miRNAs and lncRNAs in blood and saliva from boys with severe airflow obstruction (SAO) and compared those with normal lung function (NLF) to those with evidence of severe airflow obstruction.

## 2. Results

### 2.1. Clinical Differences in Pediatrics Severe Asthma

The subject recruitment process is outlined in Figure 1A. The demographic and clinical characteristics of children with controlled asthma with normal lung function (NLF) and those with severe asthma and abnormal lung function (SAO) are shown in Table 1. SAO subjects had significantly higher FVC values (3.93 ± 1.51) compared to the normal group (2.58 ± 0.43, *p* = 0.019). However, there was no significant difference in the FVC% predicted between the groups (*p* = 0.58). FEV1 was slightly higher in the SAO group (2.53 ± 1.05) compared to the NLF group (2.23 ± 0.37), but the difference was not statistically significant (*p* = 0.43). In contrast, the FEV1% predicted was significantly lower in the SAO children (82.11 ± 17.81) compared to NLF subjects (112.51 ± 9.81, *p* = 0.0003). The FEV1/FVC ratio was markedly reduced in SAO (64.52 ± 7.77) versus the NLF (86.52 ± 1.63; *p*-value < 0.0001). Similarly, the FEF75% predicted was dramatically lower in SAO (39.72 ± 10.61) than in NLF (124.74 ± 19.09, *p* = 0.0001). There were no significant differences in circulating eosinophil counts between the groups (*p* = 0.28). However, total IgE levels were significantly elevated in the SAO group (319.33 ± 214.22) compared to the NLF group (72.21 ± 62.43, *p* = 0.004). Taken together, the data indicates that SAO children exhibit marked reductions in several lung function parameters and elevated IgE levels, distinguishing them clearly from NLF children.

### 2.2. Characterization of Plasma- and Saliva-Derived Exosomes

Exosomes isolated from plasma and saliva were evaluated for morphology by transmission electron microscopy (TEM), for size by nanoparticle tracking analysis (NTA), and for their protein composition by flow cytometry (Figure 1B,C). The exosome size was confirmed by negative stain transmission electron microscopy (TEM) and their morphology showed that the typical cup-shaped feature ranged from 30 to 150 nm in diameter (Figure 1B), confirming the published results [30,31]. Indeed, exosomes are 30–120 nm in size and spherical in shape, with an observable bilayer lipid membrane (Figure 1B). To further verify the extraction of exosomes, flow cytometry was used to detect the exosomal marker CD63 in samples with or without the isolation of exosomes (Figure 1C). Exosome concentrations were measured using Nanoparticle Tracking Analysis (NTA). In plasma, exosome concentration levels were 5.93 × 10^−9^ ± 0.65 × 10^−9^ in NLF asthmatic children and 5.65 × 10^−9^ ± 0.76 × 10^−9^ in SAO children (*p* = 0.56). In saliva, the levels were 5.63 × 10^−9^ ± 0.55 × 10^−9^ in NLF and 7.32 × 10^−9^ ± 0.78 × 10^−9^ in SAO (*p* = 0.45). These results confirm the successful isolation and characterization of exosomes from plasma and saliva, with the typical morphology, size, and marker expression. Exosome concentrations did not differ significantly between NLF and SAO groups in either fluid, supporting their suitability for downstream molecular analysis.

### 2.3. Exosomal miRNA Profiling in Pediatric Plasma and Saliva

We analyzed differences in the miRNA expression in plasma and salivary exosomes between SAO and NLF. In plasma-derived exosomes, 27 differentially expressed miRNAs (DEMs) were identified (Table 2), while 40 DEMs were found in saliva-derived exosomes (Table 3). The most significant miRNAs in plasma were hsa-miR-25-3p (fold change: 1.59, *p* = 2.65 × 10^−20^) and hsa-miR-451b-5p (fold change: 6.8, *p* = 7.1 × 10^−19^). In saliva, the most significant changes were observed in hsa-miR-345-5p (fold change: 1.81, *p* = 6.87 × 10^−7^) and hsa-miR-25-3p (fold change: 2.08, *p* = 2.17 × 10^−6^). To explore the overlap between the two sample sources, a Venn diagram analysis was performed (Figure 2A), which revealed that 22 miRNAs were unique to plasma, 35 miRNAs were unique to saliva, and 5 miRNAs were shared between plasma and saliva. These findings suggest that saliva and plasma exosomes may provide complementary, yet distinct, molecular insights into asthma severity.

### 2.4. Exosomal miRNAs and Their Gene Targets in Plasma and Saliva

In plasma-derived exosomes, hsa-miR-141-3p showed the widest gene targeting, including regulators of growth and inflammation like IGF1R and MAPK9. In addition, hsa-miR-200a-3p targeted genes tied to immune signaling, such as HGF (Figure 2B). Shared miRNAs hsa-miR-16-5p and hsa-miR-122-5p linked to SOCS2 and PHLPP2, suggesting cross-fluid regulatory roles. hsa-miR-25-3p and hsa-miR-92a-3p again targeted CDH1, reinforcing their relevance in epithelial regulation. hsa-miR-486-5p targeted DOCK3, implicating cytoskeletal dynamics. Additional miRNAs like hsa-miR-451a, miR-203a-3p, and miR-27a-5p were tied to asthma-related genes including MYC, EGFR, and BMI1, highlighting their involvement in airway inflammation and remodeling.

In saliva, Figure 2C reveals a dense miRNA–gene interaction network linked to asthma, with many miRNAs targeting genes involved in immunity, inflammation, and cell cycle regulation. hsa-miR-27a-3p and hsa-miR-25-3p emerged as central hubs, regulating key genes like IFNG, TGFB2, and EZH2. Other miRNAs, including hsa-miR-16-5p, hsa-miR-27b-3p, and hsa-miR-145-5p, also converged on major immune genes such as TP53, EGFR, and SMAD3. miRNAs like hsa-miR-223-3p and hsa-miR-143-3p targeted genes tied to apoptosis and stress, while hsa-miR-486-3p and hsa-miR-320c regulated metabolic and oncogenic genes like FASN and MYC. Overall, several miRNAs, especially hsa-miR-25-3p, hsa-miR-27a-3p, and hsa-miR-16-5p, may serve as core regulators and potential therapeutic targets in asthma.

The interaction network of key miRNAs and their targets across plasma and saliva are shown in Figure 2D. hsa-miR-25-3p, hsa-miR-16-5p, and hsa-miR-92a-3p were found in plasma, while hsa-miR-486-3p was saliva-specific. Notably, hsa-miR-25-3p appeared in both, suggesting a shared regulatory role. In plasma, these miRNAs targeted genes involved in immunity, apoptosis, and cell regulation, such as ERBB2, TP53, and PTEN. In saliva, hsa-miR-486-3p targeted FASN, linked to lipid metabolism. The overlap and distinct targeting patterns suggest both shared and fluid-specific miRNA regulation in asthma.

### 2.5. KEGG Pathway Analysis of miRNA Targets in Plasma and Saliva

The Venn diagram in Figure 2A shows the number of miRNAs found in plasma (n = 22), saliva (n = 34), and those shared between both (n = 5). An enrichment analysis of 27 plasma miRNAs identified several significantly enriched KEGG pathways, with *p*-values confirming statistical relevance (Appendix A). The number of genes per pathway and targeting miRNAs were identified. The most enriched pathway was fatty acid biosynthesis (hsa00061) (*p* = 5.44 × 10^−15^), with two genes targeted by one miRNA. Focal adhesion (hsa04510) and mTOR signaling (hsa04150) followed, targeted by seven and nine miRNAs, respectively. PI3K-Akt signaling (hsa04151) is also significant (*p* = 7.20 × 10^−9^), involving 73 genes and 6 miRNAs. Highly miRNA-targeted pathways include mTOR (nine miRNAs), MAPK (eight), focal adhesion, ubiquitin-mediated proteolysis, and small-cell lung cancer (seven each), plus several cancer and signaling pathways (six each). These results suggest plasma miRNAs predominantly target cancer, signaling, and metabolic pathways, with certain pathways acting as potential disease hubs.

A pathway enrichment analysis of 40 saliva-derived miRNAs identified 107 significantly enriched KEGG pathways (Appendix A). The most enriched included PI3K-Akt signaling, MAPK signaling, actin cytoskeleton regulation, and focal adhesion (*p* = 3.93 × 10^−30^), each involving over 100 genes and 29 miRNAs. PI3K-Akt signaling was top-ranked (*p* = 3.40 × 10^−51^), with 174 genes and 31 miRNAs. MAPK and focal adhesion followed closely, with 133 and 103 genes, and up to 32 miRNAs each. Other highly enriched pathways included cancer, axon guidance, Wnt signaling, ubiquitin-mediated proteolysis, and neurotrophin signaling (*p* = 1 × 10^−20^). Cancer-related (e.g., prostate, pancreatic, and glioma) and immune-related pathways (e.g., T cell receptor and Fc gamma R-mediated phagocytosis), along with neurotransmitter-related synapse pathways, were also prominent, suggesting a regulatory role for saliva miRNAs in proliferation, signaling, immunity, and tumor progression.

### 2.6. KEGG Pathways of Exosomal miRNAs in Shared Plasma and Saliva

A comparative KEGG analysis of saliva and plasma revealed both shared and distinct pathways. KEGG pathway mapping for each group (Appendix A) showed 99 total pathways: 55 (55%) unique to saliva, 11 were unique to plasma, and 44 were shared (Figure 3A). The five shared miRNAs were enriched in 37 pathways, with top associations including PI3K-Akt (*p* = 2.11 × 10^−12^), focal adhesion (*p* = 4.44 × 10^−12^), and Wnt signaling (*p* = 2.27 × 10^−9^) spanning cancer, immunity, neurobiology, and metabolism. Notably, we identified 19 genes involved in B cell signaling (*p* = 1 × 10^−7^) and 23 in T cell signaling *(p* = 9 × 10^−7^), as shown in Figure 3B,C. Overall, saliva exhibited greater pathway diversity, but the shared core pathways highlight the complementary diagnostic potential of both biofluids in microbiome-based profiling.

### 2.7. KEGG Pathway Analysis Specific to miRNAs-Plasma and Saliva

Next, we analyzed KEGG pathways targeted by exosomal miRNAs specifically in a saliva and plasma Venn diagram (Figure 3A). Pathway enrichment revealed that plasma miRNAs were most associated with mTOR, MAPK, focal adhesion, and ErbB signaling, while saliva showed the strongest enrichment in ubiquitin-mediated proteolysis, ErbB, PI3K-Akt, and cytoskeletal regulation (Appendix A). Saliva showed broader coverage, with higher gene counts in key pathways including the regulation of the actin cytoskeleton, PI3K-Akt signaling pathway, and MAPK signaling pathway (Appendix A). Overall, plasma and saliva miRNAs target overlapping but distinct pathways, supporting their complementary value in noninvasive biomarker discovery for asthma and related conditions.

### 2.8. Comparative Analysis of lncRNA Expression in Plasma and Saliva

Differential expression analysis of lncRNAs in plasma and saliva was conducted to identify significantly altered transcripts. Volcano plots (Figure 4A) display the relationship between the statistical significance (−log_10_ *p*-value) and expression magnitude (log_2_ fold change). In plasma, 604 lncRNAs were significantly differentially expressed (log_2_ FC ≥ 2, adjusted *p* ≤ 0.05), including 100 upregulated and 503 downregulated transcripts. Log_2_ fold changes ranged from −6 to +4 (Figure 4B). The most dysregulated lncRNAs clustered in the plot’s upper corners, suggesting biomarker potential. In saliva, 115 lncRNAs showed significant changes, with 74 upregulated and 41 downregulated (log_2_ FC ≥ 2, adjusted *p* ≤ 0.05). The volcano plot revealed clear separation, indicating distinct expression patterns. A comparison between plasma and saliva uncovered both overlapping and unique lncRNA signatures, reflecting compartment-specific expression and supporting the combined use of both fluids for biomarker discovery in systemic and localized conditions. A Venn diagram (Figure 4C) illustrated the distribution of differentially expressed lncRNAs: 604 in plasma and 73 in saliva, highlighting both shared and exclusive transcripts. Lists of differentially expressed lncRNA in plasma and saliva are shown in Appendix A, respectively. These findings support the combined analysis of plasma and saliva exosomal lncRNAs for comprehensive biomarker discovery in asthma.

### 2.9. KEGG Pathways for lncRNAs in Plasma and Asthma

The NcPath tool was used to link KEGG pathway associations with changes in lncRNAs and miRNAs in plasma and saliva, identifying statistically significant interactions (Appendix A). Pathway enrichment analysis revealed several significantly enriched signaling pathways. In plasma, the top pathways included PI3K-Akt, VEGF signaling, and cellular senescence, with adjusted *p*-values as low as 4.3 × 10^−14^. The PI3K-Akt pathway had the highest gene count (143), suggesting strong involvement in the biological context. Other notable pathways were glucagon signaling, a tight junction, and multiple cancer-related pathways such as hepatocellular carcinoma, olorectal cancer, and breast cancer.

In saliva, the PI3K-Akt pathway again ranked highest, involving 185 genes with an adjusted *p*-value of 2.7 × 10^−18^ (Appendix A). Other enriched pathways included human T-cell leukemia virus 1 infection, Rap1 signaling, chemokine signaling, and focal adhesion. Immune pathways (e.g., NF-kappa B, Toll-like receptor, and T cell receptor) and cancer-related pathways (prostate, pancreatic, and gastric cancer) were also significantly represented. Metabolic and endocrine pathways—insulin, adipocytokine, and thyroid hormone signaling—along with aging-related pathways like cellular senescence and ferroptosis, were enriched as well. Together, these results point to a complex interplay of immune systems, cancer, and metabolic pathways, with PI3K-Akt signaling emerging as a central hub in both plasma and saliva datasets. We identified several pathways associated with asthma in plasma based on significantly lower adjusted *p*-values. These pathways focus on immune-related, inflammation-related, and airway remodeling/cell signaling pathways that are well-established in asthma pathophysiology.

Several canonical pathways were particularly enriched: The MAPK signaling pathway (hsa04010, *p* = 7.56 × 10^−13^)—central to cytokine signaling, inflammation, and airway remodeling, core mechanisms in asthma. The TNF signaling pathway (hsa04668, *p* = 9.59 × 10^−8^)—TNF-α is a key mediator of airway inflammation and bronchial hyper-responsiveness. Toll-like receptor signaling pathway (hsa04620, *p* = 6.07 × 10^−4^)—critical for recognizing allergens and pathogens; activates innate immune responses. IL-17 signaling pathway (hsa04657, *p* = 6.21 × 10^−4^)—IL-17-producing cells contribute to neutrophilic inflammation in SAO. Chemokine signaling pathway (hsa04062, *p* = 0.035)—drives immune cell recruitment to inflamed airways, a hallmark of asthma. These pathways underscore the multifaceted immune system and signal disruptions that underline asthma, especially in its SAO or treatment-resistant forms.

## 3. Discussion

This study aimed to identify miRNA and lncRNA biomarkers linked to asthma severity in boys with severe airflow obstruction (SAO) by comparing plasma and salivary profiles to asthmatic children with normal lung function (NLF). Using spirometry-defined criteria and small RNA sequencing from blood- and saliva-derived exosomes, we generated a comprehensive profile of circulating miRNAs and lncRNAs. Children with SAO had elevated IgE, eosinophils, and allergen sensitization, reflecting the heightened inflammation typical of severe asthma. Our analysis of exosomal miRNAs from plasma and saliva revealed distinct expression patterns between groups, with five overlapping miRNAs identified in both sample sources. These miRNAs were linked to immune pathways including neurotrophin, the T-cell receptor, and B-cell receptor signaling, suggesting a systemic signature of immune dysregulation. Given their presence in saliva, these miRNAs offer a non-invasive way to monitor asthma severity and could help guide treatment. Beyond biomarkers, they may also reveal targets for therapy. Additionally, interactions between lncRNAs and miRNAs likely influence gene expression, further shaping asthma pathophysiology.

Asthma is a chronic inflammatory airway disease influenced by both environmental and genetic factors [9]. It presents multiple phenotypes, each with distinct yet somewhat overlapping pathophysiological mechanisms and clinical features, making diagnosis and treatment challenging [32,33]. Among the two clinical groups selected for the present study, several parameters showed significant variation, reflecting the physiological burden of severe disease. Children with SAO were significantly older than those with NLF (*p* = 0.006). SAO patients showed higher FVC (*p* = 0.0197), indicating a larger lung volume; however, their FEV1% predicted (*p* = 0.00028) and FEV1/FVC ratio (*p* = 1.42 × 10^−7^) were markedly lower, reflecting a more pronounced airflow limitation. The FEF75% predicted was dramatically reduced in SAO compared to NLF (*p* = 7.84 × 10^−10^), underscoring substantial small airway dysfunction. Additionally, IgE levels were significantly higher in the SAO group (*p* = 0.01), suggesting a stronger allergic component. No significant differences were noted in the FVC% predicted (*p* = 0.579) or absolute FEV1 values (*p* = 0.427), indicating that while the overall lung size may be similar, airway obstruction specifically distinguishes the SAO group. Recent studies highlight the role of exosomes in asthma [33]. Their composition reflects their cell of origin and physiological state, containing proteins, nucleic acids, and lipids. All major immune cells release exosomes, which contribute to the development and progression of asthma and other inflammatory diseases [33]. Our study examined the expression of plasma- and saliva-derived exosomal miRNAs in severely affected asthmatic children compared to those with mild and well-controlled disease. Our aim was to determine whether circulating exosomal miRNAs from plasma or saliva or a combination of both are differentially expressed in SAO cases and whether their differential expression could provide cues to approaches aimed at further characterizing severe pediatric asthma.

Several studies have shown that miRNAs are able to act on all the principal effector cells of airway hypersensitivity, which could play a crucial role in asthma genesis [34]. miRNAs could have a decisive role in regulating the phenotypic setting of the T helper cell type 2 (Th2) and the response to neutrophils, eosinophils, macrophages, T lymphocytes, mast cells (MCs), and epithelial cells to augment the secretion of cytokines that support the increase in inflammatory alterations [34].

### 3.1. miRNA Profiling and KEGG Pathways

Emerging studies underscore the pivotal role of miRNAs in regulating immune responses and airway remodeling in asthma, particularly when delivered via engineered exosomes. At the molecular level, miRNAs exert substantial regulatory influence over immune cell differentiation relevant to asthma pathogenesis. For example, miR-493-5p inhibits Th9 cell differentiation by targeting FOXO1, a transcription factor critical to pro-inflammatory signaling pathways [35]. As Th9 cells contribute to allergic inflammation via IL-9 secretion, modulating this pathway through miRNA-based strategies presents a promising avenue for allergic asthma management. These findings are consistent with another study that highlighted miRNAs as both key mediators and potential therapeutic targets in airway hyper-reactivity following respiratory syncytial virus (RSV) infection [36].

Asthma heterogeneity, especially in SAO neutrophilic forms, remains a treatment challenge. miRNAs that regulate Th17 and innate immune responses show potential, particularly exosomal miRNAs, which act as both disease markers and immune modulators. Conversely, exosomal miRNA signatures have been identified in allergic asthma, suggesting their value as non-invasive biomarkers [37]. These findings support exosome-delivered miRNA therapies as a targeted, low-toxicity option. However, clinical use hinges on better understanding their pharmacokinetics, biodistribution, and immunogenicity. Future research should integrate miRNA diagnostics with personalized treatment strategies [38].

We show in Table 2 a panel of significantly dysregulated exosomal miRNAs in plasma, with both up- and downregulated candidates identified in SAO children by a log_2_ fold change and adjusted *p*-values. Several miRNAs, including hsa-miR-451b-5p, hsa-miR-7706, hsa-miR-195-3p, and hsa-miR-141-3p showed strong upregulation, suggesting roles in the disease process. Others like hsa-miR-3158-5p, hsa-miR-375-3p, and hsa-miR-122-5p were also highly upregulated. Even miRNAs with modest fold changes such as hsa-miR-16-2-3p, hsa-miR-16-5p, and hsa-miR-25-3p exhibited highly significant *p*-values, indicating consistent differential expression. A subset of miRNAs that was notably downregulated included hsa-miR-4665-5p, which showed the greatest decrease (log_2_ FC = −7.63, *p* = 0.03), with others like hsa-miR-203b-5p, hsa-miR-1246, and hsa-miR-125a-5p that were also significantly reduced in SAO. These patterns suggest disrupted regulatory mechanisms and point to miRNAs such as hsa-miR-451b-5p, hsa-miR-3158-5p, and hsa-miR-4665-5p as promising candidates for follow-up.

We also show in Table 3 the upregulated salivary exosomal miRNAs. For example, hsa-miR-550a-3-5p had the highest fold change, meeting the significance threshold and suggesting biological relevance. hsa-miR-486-3p, hsa-miR-3615-3p, and hsa-miR-106b-3p showed log_2_ FC > 2.2 and strong *p*-values (<0.01), reinforcing their diagnostic potential. Similarly, hsa-miR-25-3p, hsa-miR-345-5p, and hsa-miR-92a-3p showed low statistical significance and marked upregulation. Several of these miRNAs are involved in immune and inflammatory pathways, supporting their relevance to asthma-related disease mechanisms. The data support saliva as a non-invasive source for biomarker discovery.

In addition, we present in Table 4 a list of five miRNAs, hsa-miR-501-3p, hsa-miR-486-3p, hsa-miR-16-5p, hsa-miR-25-3p, and hsa-miR-92a-3p, that were significantly upregulated in both plasma and saliva, showing consistency across these two sources. For instance, hsa-miR-501-3p had a stronger fold change in plasma (log_2_ FC = 3.83) than in saliva (1.79), whereas hsa-miR-486-3p showed the opposite trend. These overlaps point to a stable, cross-fluid expression signature. Given saliva’s non-invasive nature, an especially valuable feature in pediatric or low-resource settings, these shared miRNAs hold strong promise for biomarker development. Further validation is necessary to establish their clinical utility.

### 3.2. KEGG Pathways for miRNAs’ Target Genes

Several KEGG pathways are shared between plasma and saliva in asthma, highlighting key biological processes active both systemically and locally (Table 5): (1) Immune activation: Shared pathways like PI3K-Akt, MAPK, T cell receptor, and B cell receptor signaling reflect broad immune involvement in asthma. These are central to inflammation, lymphocyte activation, and cytokine production. (2) Barriers and structural changes: pathways such as focal adhesion, actin cytoskeleton regulation, the adherent’s junction, and the gap junction point to epithelial remodeling and cell migration, core features of asthma pathophysiology. (3) Neuro-immune link: pathways involving synapses and neurotrophin signaling suggest neural influence on inflammation and airway hyper-responsiveness, supporting asthma as a neuro-immune disease. (4) Cell trafficking and clearance, endocytosis, ubiquitin proteolysis, and Fc receptor-mediated phagocytosis indicate active immune complex processing and antigen presentation. (5) Vascular and hypoxia signals in VEGF and HIF-1 pathways suggest involvement in vascular changes and hypoxia; both are tied to SAO. These overlaps validate saliva as a non-invasive fluid for tracking asthma-related molecular changes and support its use in future diagnostics.

The PI3K-Akt signaling pathway plays a crucial role in asthma pathogenesis, particularly in allergic asthma. The activation of this pathway promotes inflammation, airway remodeling, and exacerbates the disease. Inhibiting PI3K and AKT activity, either directly or by targeting downstream effectors, has shown promise in reducing asthma symptoms and inflammation [39]. miRNAs bind to their targets with partial complementarity, such that any miRNA is capable of binding hundreds or even thousands of targets. As miRNAs can regulate functionally related genes, it is possible that a few miRNAs or even a single miRNA could regulate entire pathways. Many of the differentially expressed miRNAs target pathways specific to inflammation. The role of the MAPK signaling pathway is well known in the pathogenesis of asthma [40]. Most of the miRNA groups also target the phosphoinositide 3-kinase-Akt pathway, which is another particularly well-studied signaling cascade in TH2 inflammation [41]. Focal adhesion pathways in our study are likely involved in changes in the airway basement membrane and airway smooth muscle in asthmatic patients [42].

### 3.3. lncRNAs Profiling and KEGG Pathways

lncRNAs do not produce proteins but play key roles in regulating gene expression. lncRNAs are constituted by more than 200 nucleotides and have an important role as regulators during development and pathological processes [43,44]. lncRNAs and miRNAs are major types involved in the immune response, cell proliferation, apoptosis, differentiation, polarization, and cytokine secretion. Their interactions are critical in lung inflammatory diseases and may offer therapeutic targets [45]. lncRNAs and miRNAs are the most studied classes of non-coding RNAs involved in pathological conditions [43,44]. Some lncRNAs are involved in cell–cell communication via exosomes. Due to their complex structures, they can also serve as scaffolds for proteins, mRNAs, or miRNAs, helping to load these molecules into exosomes [46].

lncRNA and miRNA bind and interact with each other in a variety of ways, influencing downstream gene expression [47]. Although miRNAs have been widely studied in asthma [48,49], the role of exosomal lncRNAs remains largely unexplored. Only about 2% of the human genome consists of protein-coding genes. The remaining 98% comprises noncoding RNAs, including ribosomal RNA (rRNA), transfer RNA (tRNA), ribozymes, small nuclear RNA (snRNA), small nucleolar RNA (snoRNA), miRNAs, lncRNAs, and circular RNAs (circRNAs) [47]. MiRNAs and long lncRNAs regulate key biological processes and their abnormal expression may contribute to diseases like asthma.

lncRNAs, though less studied due to low expression and poor sequence conservation, are key regulators of gene expression. These non-protein-coding RNAs (>200 nucleotides) influence DNA methylation, histone modification, chromatin remodeling, and transcriptional activity. They can act as decoys, blocking miRNA activity and restoring target gene expression, or compete with miRNAs for mRNA binding sites to prevent repression [50].

In childhood asthma, lncRNAs are increasingly recognized as drivers of disease progression and promising targets for diagnosis and treatment. They influence inflammation, airway remodeling, and immune regulation by interacting with miRNAs and related pathways [51,52]. Several mechanistic studies have further elucidated the functional significance of specific lncRNA–miRNA–mRNA interactions. For example, it has been demonstrated that silencing lncRNA ANRIL alleviates airway remodeling via the miR-7-5p/EGR3 pathway, indicating a promising therapeutic axis [53]. In parallel, Wang and Chen found that lncRNA TUG1 promotes the proliferation and migration of airway smooth muscle cells (ASMCs) through the regulation of the miR-216a-3p/SMURF2 axis, linking it to structural airway changes seen in chronic asthma [54]. LINC-PINT regulates immune responses in asthma by inhibiting abnormal ASMC proliferation via the miR-26a-5p/PTEN pathway, indicating a tumor suppressor-like function [55]. Similarly, the knockdown of lncRNA NORAD reduces airway remodeling by modulating the miR-410-3p/RCC2 axis and inhibiting Wnt/β-catenin signaling, highlighting its potential as a therapeutic target for chronic airway changes [56].

The diagnostic and prognostic value of lncRNAs is gaining traction, with the PTTG3P/miR-192-3p/CCNB1 axis emerging as a potential biomarker for childhood asthma, underscoring their promise in non-invasive disease monitoring [57]. lncRNA CRNDE is associated with elevated inflammatory cytokines and altered miRNA profiles in asthmatic children, reinforcing its potential clinical utility in lncRNA-based assessments [58]. lnc-NEAT1 drives the inflammatory cytokine release and is linked to a higher risk of severe asthma exacerbations [59]. KEGG pathways linked to plasma lncRNA-based adjusted *p*-values are shown Appendix A. Plasma lncRNAs influence key pathways in signal transduction (mTOR, PI3K-Akt, MAPK, and FoxO), immunity and inflammation (TNF, Toll-like, and T cell receptor), and the cell structure (adherents junctions, focal adhesion, and actin cytoskeleton), affecting metabolism, growth, apoptosis, and immune responses. This broad involvement, especially in cancer and immunity, highlights their potential as biomarkers or therapeutic targets. Saliva-derived lncRNAs show similar associations with signaling, cancer, immunity, metabolism, and structural pathways, underscoring their wide biological relevance (Appendix A). Saliva lncRNAs are linked to key biological functions in cancer, immunity, and metabolism. Top enriched pathways include cellular senescence, MAPK signaling, and the cell cycle. Core signaling networks (MAPK, PI3K-Akt, mTOR, TNF, and TGF-beta) highlight regulatory roles in inflammation and homeostasis. Immune-related pathways (Toll-like, T/B cell receptor, and IL-17) point to immune modulation. Structural roles are indicated by involvement in focal adhesion, tight junctions, and cytoskeleton regulation. Neurological pathways (axon guidance and neurotrophin signaling) suggest potential in neuroimmune and sensory functions (Appendix A).

This study focused solely on male participants, based on evidence that boys face a higher risk of severe asthma and are nearly twice as likely as girls to require emergent care. Early detection remains critical and non-invasive, saliva-based exosome profiling could offer a practical tool for primary care providers to identify and monitor asthma severity in children. However, the small sample size and sex-specific focus are key limitations. The decision to include only boys was intended to reduce biological variability and improve signal clarity in this pilot study and the funding of this study. Yet, this approach limits the generalizability of our findings. The asthma pathophysiology and exosomal RNA expressions may differ between sexes, especially across developmental stages. To fully understand the diagnostic and prognostic value of exosomal miRNAs and lncRNAs, future studies must include adequately powered, sex-balanced cohorts. Such efforts are essential to explore potential gender-specific expression patterns, validate the biomarker performance, and assess their utility in stratifying the disease severity and predicting clinical outcomes in diverse pediatric populations.

## 4. Materials and Methods

### 4.1. Subjects

Twenty school-aged males aged 9–18 years with asthma were recruited from an outpatient pediatric pulmonary clinic during routine asthma check-ups that include spirometry. This study was approved by the Institutional Review Board (IRB), protocol number # 2079022 at The University of Missouri with parents or caretakers signing informed consent and children providing their assent. Half of the children (n = 10) were tested immediately after demonstrating normal spirometry (FEV1/FVC ratio > 0.84 and FEF75 > 69% predicted). The other half (n = 10) were consented and tested on the day their spirometry results showed evidence of severe airflow obstruction (FEV1/FVC ratio < 0.70 and FEF75 < 50% predicted) [60].

### 4.2. Saliva and Blood Collection

Prior to sampling, subjects were instructed to avoid eating, drinking, and brushing their teeth for at least 30 min. The whole saliva (10 mL) was collected into 50 mL polystyrene tubes, immediately kept on ice and centrifuged at 2000× *g* for 10 min at 4 °C to remove cells and large debris. The resulting supernatants were stored at −80 °C until analysis. Following saliva collection, blood was drawn from each child in EDTA tubes, transported to the lab within 30 min, and centrifuged at 3000× *g* for 10 min. Plasma was aliquoted into 0.5 mL centrifuge tubes and stored at −80 °C. 

### 4.3. Exosome Isolations and Characterization

Plasma exosomes were isolated using a Total Exosome Isolation Kit (from plasma, Life Technologies, Marietta, OH, USA) [30,31,61]. Briefly, saliva was thawed at 37 °C and further centrifuged in 2000× *g* for 10 min at room temperature. Saliva exosomes were isolated from 10 mL of saliva using Total Exosome Isolation Reagent (from other body fluids, Life Technologies, Marietta, Carlsbad, CA, USA) and samples were mixed and incubated with 5 mL of total exosome isolation reagent and further incubated at 4 °C for 75 min. The samples were centrifuged at 10,000× *g* at 4 °C for 75 min. Pellets were suspended in 600 µL 1× PBS and samples were stored at −20 °C. The isolated exosomes were subsequently quantified and characterized following MISEV2018 guidelines [62]. 

Transmission electron microscopy (TEM) was used to determine the exosome size as previously described [30,31]. Exosomes were fixed in 2% paraformaldehyde; 5 μL of the exosome suspension was then applied to each formvar/carbon-coated 200 mesh nickel grid and allowed to adsorb for 2 min. Grids were incubated with 30 μL drops of 2% uranyl acetate and examined by electron microscopy [30]. The samples were washed with distilled water seven times (2 min each) and then they were viewed under a FEI Tecnai F30 Twin (Atlanta, GA, USA) transmission EM [30]. Exosome quantifications were determined using both NanoSight, NS300, (Malvern Panalytical, Malvern, UK) equipped with a high-sensitivity sCMOS camera, 531 nm laser, and automatic syringe pump [30,31]. Also, the number of exosomes were measured with a nCS1 instrument (Spectradyne, LLC, Signal Hill, CA, USA). Plasma or saliva samples were diluted 1000-fold with 1% Tween 20 (*v*/*v*) in DPBS and measured with TS-400 cartridges [63].

### 4.4. Small RNA-Sequencing for Plasma and Saliva

Total RNA, including miRNAs from exosomes derived from plasma or saliva, were isolated using the miRNeasy Serum/Plasma Mini Kit (Qiagen, Valencia, CA, USA), following the manufacturer’s instructions. The RNA quality and integrity were assessed with the Eukaryote Total RNA Nano 6000 LabChip assay on the Agilent 2100 Bioanalyzer (Santa Clara, CA, USA). The miRNA quality was evaluated using the Agilent Small RNA Kit. We constructed a miRNA library using the NEBNext^®^ Multiplex Small RNA Library Prep Set for Illumina (NEB, Ipswich, MA, USA) following the manufacturer’s instructions. Briefly, the total RNA, including miRNAs (100 ng), was used for library preparation. The 3′ and 5′ RNA adapters were ligated to the RNA using T4 RNA ligase, followed by reverse transcription to generate cDNA. The cDNA was then amplified via 13 cycles of PCR. To select the desired cDNA library size, PCR products were resolved on a 6% TBE gel alongside a custom ladder. Small RNA fragments (140–160 base pairs) were excised, incubated overnight, and eluted using a spin column. The resulting libraries were sequenced on the Illumina NovaSeq platform using 50-nucleotide single reads, generating 20 million reads per sample. Sequence quality control (FASTQC) was performed, followed by alignment to the human genome (GRCm39) using miRDeep2 v0.0.8 (Bowtie algorithm). Reads with a Phred score < 20 were trimmed using SolexaQA++ v3.1.7.1. (Burrows–Wheeler Aligner trimming algorithm). Reads shorter than 17 bases were discarded. For miRNA analysis, known miRNAs from miRBase (release 22) were used as reference sequences. Small RNA annotation was performed using the Unitas pipeline to remove non-miRNA sequences. mirPRo and miRDeep2 were used to quantify known miRNAs and predict putative ones. Reads were collapsed into a set of unique sequences and aligned to the ensemble human genome and miRNA sequences from miRBase (release 22).

### 4.5. Small RNA-Seq Data Processing and Analysis

The initial quality control of small RNA sequencing data was conducted with Cutadapt (v4.6), which removed the first three nucleotides, poly-A sequences, Illumina adapters, ambiguous bases (N’s), reads shorter than 20 bp, and those with a Phred quality score below 20. Paired-end reads were then concatenated for microRNA profiling using miRge3 [64], which included an error correction module to compute raw read counts. The differential expression of miRNAs was analyzed with DESeq2 [65], using a significance threshold of *p* < 0.05 and an absolute fold change ≥2. The functional enrichment of predicted miRNA target genes was performed with ClueGO (v2.5.10) [66] within Cytoscape (v3.10.3). miRge3.0 was used to explore potential miRNA. Long non-coding RNAs (lncRNAs) were annotated using GENCODE (GRCh38) [67] and RNACentral GFF3 files [68]. Reads were aligned with Bowtie [69] and raw counts were obtained via featureCounts [70]. DESeq2 was again used for differential expression analysis (adjusted *p* < 0.05; |log_2_ FC| ≥ 2). LncRNAs were further annotated using lncPedia [71] and lncBook [72] via RNACentral IDs. To investigate interactions between altered miRNAs and lncRNAs in asthma samples, NcPath [73] was used for pathway mapping based on KEGG [74]. Interaction networks were visualized in Cytoscape [75] using igraph in R (version 2.1.4) and annotated with RCy3 [76].

### 4.6. Illumina RNA Library Preparation and Sequencing

RNA libraries were prepared using the Illumina Stranded mRNA Library Prep Kit, following the manufacturer’s instructions, and the Genomics Technology Core Facility at the University of Missouri performed the small RNA sequencing analysis DNA Core. Briefly, mRNA was enriched using poly-T oligo beads, fragmented, and reverse transcribed using random hexamers. After end repair, adapters were ligated and the fragments amplified by PCR. Libraries were constructed using the manufacturer’s protocol with reagents supplied in the SMARTer small RNA library preparation kit (Takara Bio USA, Inc., San Jose, CA, USA). The library quality was assessed with an Agilent Fragment Analyzer and concentrations were measured with a Qubit fluorometer using the dsDNA HS Assay Kit (Thermo Fisher Scientific, Cincinnati, OH, USA). Pooled libraries were sequenced using paired-end 100 bp reads on an S4 flow cell, generating ~100 million paired reads per sample. Raw reads were quality checked using FastQC (v0.11.9) and summarized via MultiQC (v1.11) [77]. Adapter trimming and quality filtering (<20 bp or Phred < 20) were carried out with Trim Galore (v0.6.7), which wraps Cutadapt. High-quality reads were aligned to GRCh38.108 using HISAT2 (v2.2.1) [78], achieving ~95% alignment. The resulting BAM files were sorted with Samtools (v1.14) [79] and expression quantification was carried out using Cufflinks (v2.2.1) [80].

### 4.7. Differential Gene Expression Analysis (DGEA)

DESeq2 was used to identify differentially expressed genes (DEGs) between severe asthma and control samples in plasma and saliva. Genes with a |fold change| ≥ 2 and q-value ≤ 0.05 were considered significant. Visualizations, including volcano plots and PCA were generated using Enhanced Volcano (v1.16.0) and the princomp function in R (v4.2.2).

### 4.8. Protein–Protein Interaction (PPI) Analysis

Protein–protein interaction networks were constructed from DEGs using STRING [81]. Interaction files were imported into Cytoscape and analyzed with the CytoHubba plugin (v0.1) [81]. Hub genes were ranked using the Maximal Clique Centrality (MCC) algorithm with default parameters.

### 4.9. Statistical Analysis

Statistical analyses were conducted using R software (version 3.3.1; R: A Language and Environment for Statistical Computing) and GraphPad Prism (version 10.4.1; GraphPad Software Inc., San Diego, CA, USA). ANOVA was used for overall group comparisons, followed by the Kruskal–Wallis test for multiple comparisons involving more than two groups. For comparisons between two groups, the Mann–Whitney U test was applied. In cases where repeated measurements were taken from the same group of children, a two-way ANOVA followed by Tukey’s multiple comparisons test was used. Data are presented as mean ± SD, as indicated in the figure legends. A *p*-value ≤ 0.05 was considered statistically significant.

## 5. Conclusions

Exosomal miRNAs play critical roles in regulating inflammation and airway remodeling in asthma, highlighting their promise as both diagnostic and therapeutic tools. Their inherent stability, accessibility, and disease-specific expression profiles support their potential as non-invasive biomarkers. Therapeutically, modulating exosomal miRNA activity may enable more targeted and personalized treatment approaches. Despite this promise, several challenges remain, including the need to standardize isolation protocols, better understand miRNA sorting mechanisms, and mitigate off-target effects. Future research should aim on refining their application as prognostic biomarkers, uncovering molecular mechanisms underlying clinical phenotypes, and advancing their use in clinical trials. Combining exosomal miRNAs with existing therapies—or emerging tools like gene editing—could improve asthma management and treatment outcomes. The plasma- and saliva-derived panels of exosomal miRNAs hold great potential as a liquid biopsy for discrimination between severe and mild presentations among asthmatic children. Exosomal miRNAs from both biofluids represent a promising tool for future biomarker studies, emphasizing the possibility to substitute plasma with less-invasive saliva collection.

## Figures and Tables

**Figure 1 ijms-26-07043-f001:**
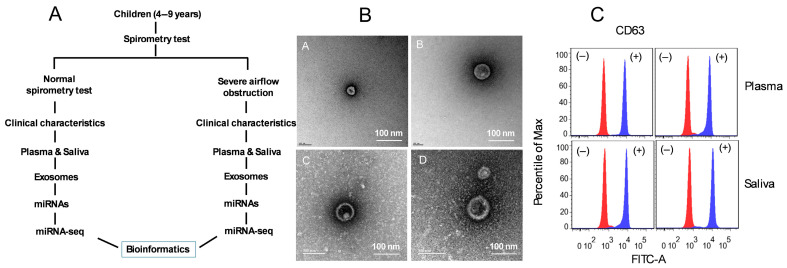
Subject recruitment and exosomes isolation and characterization. (**A**) Schema illustrate subject recruitment for asthmatic children with SAO and NLF, n = 10/group. (**B**) Transmission Electron Microscopy (TEM) images of plasma and saliva exosomes illustrating their morphology. (**B**) NanoSight Tracking Analysis (NTA) results showing the size distribution and concentration of exosomes. (**C**) Flow cytometry analysis of purified exosomes using magnetic beads coated with anti-CD63 to demonstrate the presence of exosome in the samples, n = 6/group. Scale bar is 100 µm.

**Figure 2 ijms-26-07043-f002:**
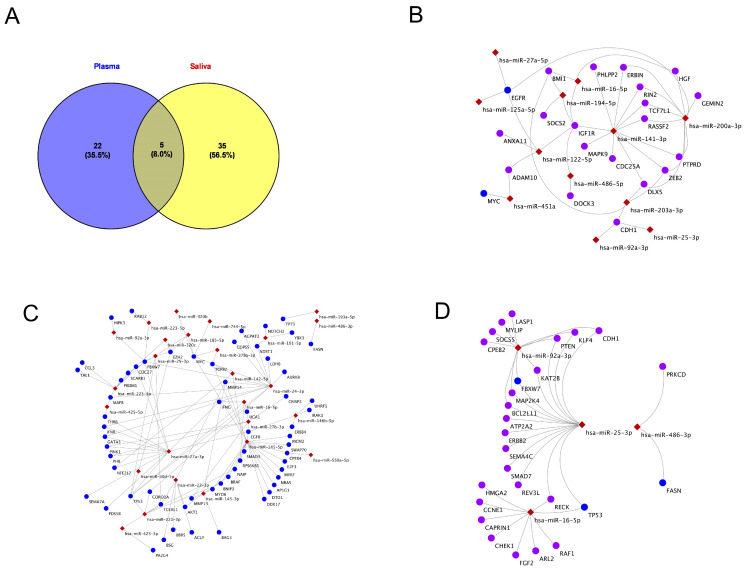
Venn diagram and predicted miRNA–target gene interactions in plasma and saliva. (**A**) Venn diagram illustrates the distribution of miRNAs detected in plasma and saliva samples. A total of 22 miRNAs were unique to plasma, 35 miRNAs were unique to saliva, and 5 miRNAs were shared between both biofluids. (**B**) Network visualization of predicted gene targets regulated by plasma-derived miRNAs. Nodes represent miRNAs and their predicted target genes, with edges indicating computationally inferred regulatory interactions. (**C**) miRNA–gene interaction network based on saliva-derived miRNAs, highlighting both unique and overlapping gene targets in comparison to plasma. (**D**) miRNA–gene interaction network derived from miRNAs common to both plasma and saliva samples. Each panel reveals the complexity and potential tissue-specific roles of circulating miRNAs in gene regulation.

**Figure 3 ijms-26-07043-f003:**
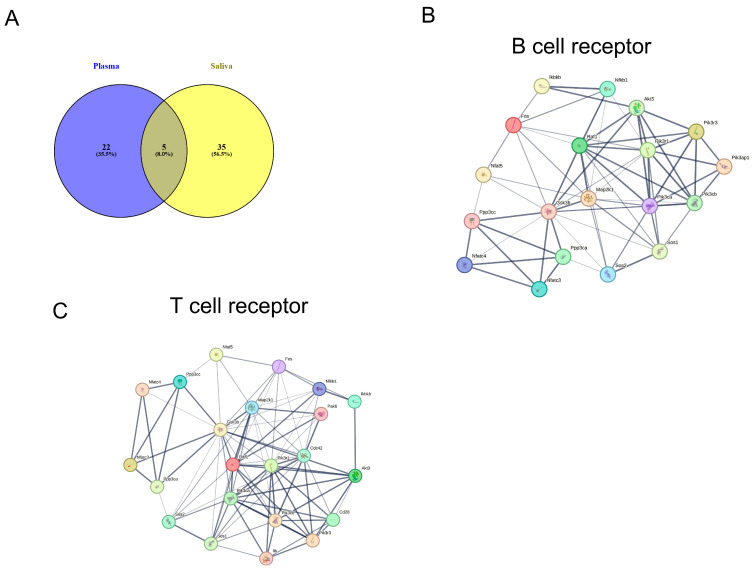
KEGG pathways in target genes miRNAs in specific plasma and saliva. Panel (**A**) shows Venn diagram between KEGG pathways in specific plasma and saliva. Panel (**B**) is pathway network for B cell receptors pathway and panel (**C**) for T cell receptors pathway.

**Figure 4 ijms-26-07043-f004:**
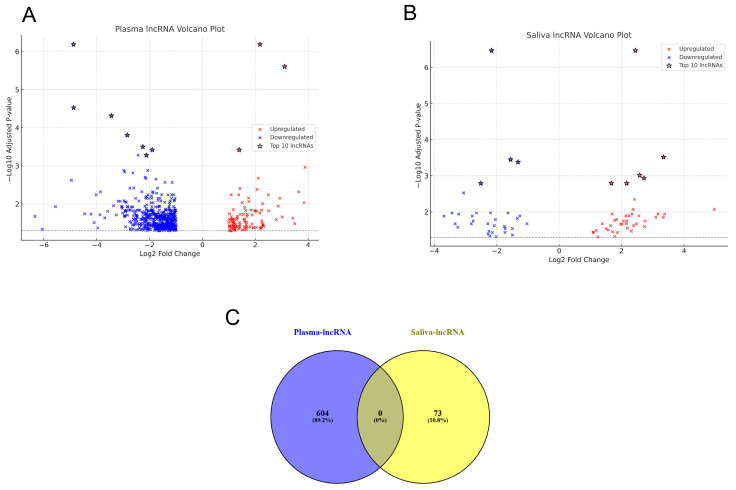
Volcano plot analysis of plasma and saliva lncRNAs in SAO vs. NLF. Differential expression analysis of long non-coding RNAs (lncRNAs) was performed using volcano plots to compare severe cases to healthy controls. (**A**) shows the differential expression analysis of lncRNAs in plasma, while (**B**) shows the differential expression analysis of lncRNAs in saliva. The plots display log_2_ fold changes versus statistical significance (−log_10_ *p*-value) for plasma-derived lncRNAs and for saliva-derived lncRNAs collected from the same pediatric subjects. Significantly upregulated (red color) and downregulated lncRNAs are highlighted (blue color), providing insight into potential biomarkers and regulatory mechanisms specific to severe disease states in both biofluids. (**C**) shows Venn diagram for the differentially expressed lncRNAs in plasma and saliva.

**Table 1 ijms-26-07043-t001:** The demographic and clinical characteristics of asthmatic children.

Items	Normal Lung Function (NLF)	Severe Airflow Obstruction (SAO)	*p*-Value
Age	10.71 ± 1.33	13.81 ± 2.67	0.006
FVC	2.58 ± 0.43	3.93 ± 1.51	0.01973155
FVC% Pred	112.51 ± 10.48	108.91 ± 16.01	0.579436397
FEV1	2.23 ± 0.37	2.53 ± 1.05	0.427157081
FEV1% Pred	112.51 ± 9.81	82.11 ± 17.81	0.000287177
FEV1/FVC Ratio	86.52 ± 1.63	64.52 ± 7.77	1.42353 × 10^−7^
FEF75% Pred	124.74 ± 19.09	39.72 ± 10.61	7.83717 × 10^−10^
IgE	72.11 ± 62.54	288.12 ± 223.19	0.01

**Table 2 ijms-26-07043-t002:** List of significant miRNAs in plasma.

Name	Log_2_ Fold Change	*p*-Value	Adjusted *p*-Value
hsa-miR-451b-5p	6.873	<0.0001	<0.0001
hsa-miR-7706	5.907	0.003	0.02
hsa-miR-195-3p	5.825	<0.0001	0.0004
hsa-miR-141-3p	5.218	0.004	0.03
hsa-miR-3158-5p	4.886	<0.0001	<0.0001
hsa-miR-3158-3p	4.584	<0.0001	<0.0001
hsa-miR-375-3p	4.126	<0.0001	<0.0001
hsa-miR-501-3p	3.835	<0.0001	<0.0001
hsa-miR-200a-3p	3.760	<0.0001	<0.0001
hsa-miR-451a	2.626	<0.0001	<0.0001
hsa-miR-122-5p	1.985	<0.0001	<0.0001
hsa-miR-192-5p	1.961	<0.0001	<0.0001
hsa-miR-122b-3p	1.925	<0.0001	<0.0001
hsa-miR-16-2-3p	1.851	<0.0001	<0.0001
hsa-miR-194-5p	1.653	0.0001	0.001
hsa-miR-486-3p	1.627	<0.0001	0.0001
hsa-miR-16-5p	1.601	<0.0001	<0.0001
hsa-miR-25-3p	1.594	<0.0001	<0.0001
hsa-miR-486-5p	1.525	<0.0001	0.0002
hsa-miR-92a-3p	1.068	<0.0001	<0.0001
hsa-miR-27a-5p	−1.131	0.001	0.01
hsa-miR-125a-5p	−1.168	<0.0001	0.0006
hsa-miR-203a-3p	−1.384	0.0006	0.006
hsa-miR-4433a-3p	−1.662	0.003	0.02
hsa-miR-203b-5p	−1.705	<0.0001	0.0006
hsa-miR-203b-5p	−1.705	<0.0001	0.0006
hsa-miR-1246	−1.746	0.0001	0.001
hsa-miR-4665-5p	−7.627	0.003	0.02

**Table 3 ijms-26-07043-t003:** List of significant miRNAs in saliva.

MiRNA	Log_2_ Fold Change	*p*-Value	Adjusted *p*-Value
hsa-miR-550a-3-5p	3.063	0.009	0.04
hsa-miR-486-3p	2.363	0.0009	0.007
hsa-miR-3615-3p	2.261	<0.0001	0.0003
hsa-miR-106b-3p	2.234	0.0007	0.0055
hsa-miR-25-3p	2.084	<0.0001	0.0001
hsa-miR-140-3p	2.004	<0.0001	0.0001
hsa-miR-423-3p	1.977	<0.0001	0.0003
hsa-miR-3184-5p	1.963	<0.0001	0.0003
hsa-miR-629-5p	1.909	<0.0001	0.0001
hsa-miR-223-3p	1.880	0.0001	0.001
hsa-miR-92a-3p	1.837	<0.0001	0.0001
hsa-miR-142-5p	1.818	<0.0001	0.0001
hsa-miR-345-5p	1.812	<0.0001	0.0001
hsa-miR-501-3p	1.794	0.006	0.03
hsa-miR-425-5p	1.751	<0.0001	0.0001
hsa-miR-223-5p	1.629	<0.0001	0.0002
hsa-miR-146b-5p	1.624	<0.0001	0.0007
hsa-let-7d-3p	1.618	<0.0001	0.0004
hsa-miR-24-3p	1.609	<0.0001	0.0001
hsa-miR-140-5p	1.578	0.0008	0.006
hsa-miR-199a-3p	1.559	0.0003	0.003
hsa-miR-744-5p	1.552	0.002	0.01
hsa-miR-221-3p	1.541	<0.0001	0.0002
hsa-miR-145-5p	1.528	0.004	0.02
hsa-miR-3074-5p	1.512	<0.0001	0.0001
hsa-miR-23a-3p	1.483	0.0002	0.002
hsa-miR-143-3p	1.478	0.0004	0.003
hsa-miR-193a-5p	1.447	0.0001	0.001
hsa-miR-941	1.373	0.001	0.01
hsa-miR-378a-3p	1.340	<0.0001	0.0003
hsa-miR-1307-3p	1.298	0.0007	0.005
hsa-miR-320a-3p	1.279	0.004	0.02
hsa-miR-652-3p	1.273	0.002	0.01
hsa-miR-191-5p	1.248	0.003	0.02
hsa-miR-185-5p	1.235	0.002	0.01
hsa-miR-16-5p	1.197	0.004	0.02
hsa-miR-27a-3p	1.150	0.001	0.007
hsa-miR-22-3p	1.084	0.0001	0.001
hsa-miR-30d-5p	1.025	0.003	0.01
hsa-miR-103b	1.015	0.01	0.04

**Table 4 ijms-26-07043-t004:** List of common significant miRNAs in plasma and saliva.

Plasma		Saliva	
Name	Log_2_ Fold Change	Adjusted *p*-Value	Log_2_ Fold Change
hsa-miR-501-3p	3.83	1.57 × 10^−5^	1.79
hsa-miR-486-3p	1.62	0.0001	2.36
hsa-miR-16-5p	1.60	1.83 × 10^−15^	1.19
hsa-miR-25-3p	1.59	9.64 × 10^−18^	2.08
hsa-miR-92a-3p	1.067	3.15 × 10^−7^	1.83

**Table 5 ijms-26-07043-t005:** List of KEGG pathways that were identified in common miRNAs bewteen plasma and saliva.

KEGG Pathway	KEGG IDs	*p*-Value	#Genes	#miRNAs
PI3K-Akt signaling pathway	hsa04151	2.11 × 10^−12^	61	5
Focal adhesion	hsa04510	4.44 × 10^−12^	43	5
Wnt signaling pathway	hsa04310	2.27 × 10^−9^	35	5
Non-small cell lung cancer	hsa05223	7.41 × 10^−8^	15	5
Regulation of actin cytoskeleton	hsa04810	1.36 × 10^−7^	39	5
B cell receptor signaling pathway	hsa04662	1.48 × 10^−7^	19	5
Small cell lung cancer	hsa05222	6.48 × 10^−7^	19	5
Insulin signaling pathway	hsa04910	8.42 × 10^−7^	27	5
Neurotrophin signaling pathway	hsa04722	9.18 × 10^−7^	25	5
T cell receptor signaling pathway	hsa04660	9.23 × 10^−7^	23	5
MAPK signaling pathway	hsa04010	2.31 × 10^−6^	44	5
Acute myeloid leukemia	hsa05221	3.98 × 10^−6^	15	5
Long-term potentiation	hsa04720	5.06 × 10^−6^	16	5
Dopaminergic synapse	hsa04728	9.10 × 10^−6^	26	5
Ubiquitin mediated proteolysis	hsa04120	9.52 × 10^−6^	26	5
Glioma	hsa05214	1.19 × 10^−5^	16	5
VEGF signaling pathway	hsa04370	1.59 × 10^−5^	15	5
Chronic myeloid leukemia	hsa05220	0.00017042	16	5
Type II diabetes mellitus	hsa04930	0.000395913	11	5
Renal cell carcinoma	hsa05211	0.000395913	16	5
Endocytosis	hsa04144	0.000453408	32	5
ErbB signaling pathway	hsa04012	0.000758374	16	5
Fc gamma R-mediated phagocytosis	hsa04666	0.001040455	17	5
Axon guidance	hsa04360	0.001191295	24	5
HIF-1 signaling pathway	hsa04066	0.002165998	19	5
Cholinergic synapse	hsa04725	0.002246121	21	5
Bacterial invasion of epithelial cells	hsa05100	0.004172215	13	5
Calcium signaling pathway	hsa04020	0.00556725	27	5
Gap junction	hsa04540	0.00633416	16	5
Glutamatergic synapse	hsa04724	0.007451913	19	5
Phosphatidylinositol signaling system	hsa04070	0.008247675	14	5
GnRH signaling pathway	hsa04912	0.009589913	15	5
Viral carcinogenesis	hsa05203	0.01249035	25	5
Gastric acid secretion	hsa04971	0.02177694	12	5
Adherens junction	hsa04520	0.02235759	14	5
Melanogenesis	hsa04916	0.02235759	16	5
Osteoclast differentiation	hsa04380	0.02479279	19	5
Cell cycle	hsa04110	0.03556233	21	5

## Data Availability

The original contributions presented in this study are included in the article. Further inquiries can be directed to the corresponding author.

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
