# Peer review of "Childhood Asthma Biomarkers Derived from Plasma and Saliva Exosomal miRNAs"

_ijms, 2025, doi:10.3390/ijms26157043_

Round 1
Reviewer 1 Report
Comments and Suggestions for Authors
This manuscript looks at the levels of exosomal miRNA and lncRNA as possible indicators of how severe childhood asthma is, by comparing blood and saliva samples from boys with serious breathing problems and those with normal lung function. The study is timely, relevant, and well-structured. The rationale is clear, and the multi-omics approach is appropriate. However, some areas need clarification and language improvement to enhance readability and scientific results.
The results are comprehensive but at times dense. Could you summarize the key findings at the end of each major subsection for better flow?
The discussion would benefit from a stronger critical appraisal of the sample size limitations and sex-specific focus on boys. How might this affect generalizability?
Several sentences are overly long and could be simplified for clarity. For example, break up multi-clause sentences in the introduction.
Consistency in abbreviations: e.g., ensure SAO, NLF, miRNA, lncRNA are defined clearly at first use.
Figures: Ensure all figure legends are fully self-explanatory and include statistical tests where applicable.
Careful proofreading is recommended to correct minor grammatical issues.
Best.
Comments on the Quality of English LanguageCareful proofreading is recommended to correct minor grammatical issues.
Author Response
We extend our sincere gratitude to the Reviewer for his/her valuable insights and constructive feedback regarding our manuscript (Ms. No.: ijms-3759341), titled “Childhood Asthma Biomarkers Derived from Plasma and Saliva Exosomal miRNAs”. We have carefully addressed each of the comments provided by the Reviewer and have revised the manuscript accordingly. Our responses to the comments are detailed below and highlighted in red throughout the revised manuscript for ease of reference.
Reviewer 1:
This manuscript looks at the levels of exosomal miRNA and lncRNA as possible indicators of how severe childhood asthma is, by comparing blood and saliva samples from boys with serious breathing problems and those with normal lung function. The study is timely, relevant, and well-structured. The rationale is clear, and the multi-omics approach is appropriate. However, some areas need clarification and language improvement to enhance readability and scientific results.
Answer: We thank the reviewer for the thoughtful and constructive feedback. We appreciate your positive remarks on the study’s rationale, structure, and relevance, as well as the appropriateness of the multi-omics approach.
Question: The results are comprehensive but at times dense. Could you summarize the key findings at the end of each major subsection for better flow?
Answer: We thank the reviewer for their suggestion and have added a summary statement to each results section accordingly.
Question: The discussion would benefit from a stronger critical appraisal of the sample size limitations and sex-specific focus on boys. How might this affect generalizability?
Answer: We edited and justified sample size. This study focused solely on male participants, based on evidence that boys face a higher risk of severe asthma and are nearly twice as likely as girls to require emergent care. Early detection remains critical, and non-invasive, saliva-based exosome profiling could offer a practical tool for primary care providers to identify and monitor asthma severity in children. However, the small sample size and sex-specific focus are key limitations. The decision to include only boys was intended to reduce biological variability and improve signal clarity in this pilot study and funding of this study. Yet, this approach limits the generalizability of our findings. Asthma pathophysiology and exosomal RNA expressions may differ between sexes, especially across developmental stages. To fully understand the diagnostic and prognostic value of exosomal miRNAs and lncRNAs, future studies must include adequately powered, sex-balanced cohorts. Such efforts are essential to explore potential gender-specific expression patterns, validate biomarker performance, and assess their utility in stratifying disease severity and predicting clinical outcomes in diverse pediatric populations.
Question: Several sentences are overly long and could be simplified for clarity. For example, break up multi-clause sentences in the introduction.
Answer: We thank the reviewer for the suggestion. We have edited and shortened many sentences in the Introduction.
Question: Consistency in abbreviations: e.g., ensure SAO, NLF, miRNA, lncRNA are defined clearly at first use.
Answer: We included the abbreviations to be consistent. SAO (severe airflow obstruction), NLF (Normal Lung Fibroblasts), miRNA (microRNA), lncRNA (long non-coding RNA).
Figures: Ensure all figure legends are fully self-explanatory and include statistical tests where applicable.
Answer: We edited the figure legends.
Reviewer 2 Report
Comments and Suggestions for Authors
Dear authors,
Thank you for the interesting study.
However, some of the issues should be addressed. The manuscript must be submitted in the format provided in the information for authors.
- The term "flare-ups" is not suitable; "exacerbation" should be used instead.
- Extracellular vesicles are not the only way for cells to communicate, but one of the possible methods, alongside other types of communication.
- Why only boys were included in the study - later in life, in adulthood, fewer men and more women are affected by asthma. In my opinion, to discriminate against girls nowadays is not fair. It would be interesting to add girls to separate groups and compare, as it would be interesting why more boys in childhood and teenage years are affected than girls, but women are more affected by severe asthma than men. This can be addressed in the introduction, as the abstract is well-presented, but it raises questions about the main idea.
- What kind of tools were used to explore potential miRNA?
- An interesting part is that there are two possible groups based not on the markers and traditional asthma phenotyping/endotyping. Also, later in the manuscript, it is mentioned that the SAO group fitted the profile of eosinophilic or allergic asthma, while the other group fitted what? Paucigranulocytic asthma? Please present all data associated with the study subjects in the table, as it is unclear regarding CBC results, IgE levels, age, and the treatment they received.
- Lines 59-60 contain the same information.
- Interesting presentation of the results - in some cases, the data is presented in the middle of the sentence, even though there are no statistically significant differences. In other cases, it is mentioned that there are no differences, and the discussion ends. It would be ideal to review the results one more time and format them uniformly.
- In lines 119-121, it is mentioned that the results fit the previous publications - can you highlight the novelty of the study in a separate segment? Additionally, this sentence would fit perfectly in the discussion section.
- P-values are often confusing and not presented evenly. Perhaps the exact p-value is not necessary to mention with -20 degrees or other values, as it is sufficient to note that p < 0.0001.
- The highlights in the tables where the significant differences are found could be in the tables- just a suggestion
- Discussion and results presented well - no comments
- the 5.2 and 5.3 parts should be edited as there isn o conssitancy - "lab", sometimes min, sometimes minutes, it is enough to mention "manufacturer's protocol used" if it is standard, no need to mention and cite different publications; sometimes x, "times" is used; also, please revise the minutes used for centrifugation and use previously mentioned abbereviations - TEM in 463-464.
Author Response
We extend our sincere gratitude to the Reviewer for his/her valuable insights and constructive feedback regarding our manuscript (Ms. No.: ijms-3759341), titled “Childhood Asthma Biomarkers Derived from Plasma and Saliva Exosomal miRNAs”. We have carefully addressed each of the comments provided by the Reviewer and have revised the manuscript accordingly. Our responses to the comments are detailed below and highlighted in red throughout the revised manuscript for ease of reference.
Reviewer 2
However, some of the issues should be addressed. The manuscript must be submitted in the format provided in the information for authors.
- The term "flare-ups" is not suitable; "exacerbation" should be used instead.
Answer: Corrected
- Extracellular vesicles are not the only way for cells to communicate, but one of the possible methods, alongside other types of communication.
Answer: We changed as suggested.
- Why only boys were included in the study - later in life, in adulthood, fewer men and more women are affected by asthma. In my opinion, discriminating against girls nowadays is not fair. It would be interesting to add girls to separate groups and compare, as it would be interesting why more boys in childhood and teenage years are affected than girls, but women are more affected by severe asthma than men. This can be addressed in the introduction, as the abstract is well presented, but it raises questions about the main idea.
Answer: The decision to include only boys in the study was based on established epidemiological patterns showing that asthma is more prevalent and often more severe in males during childhood and adolescence. Focusing on a single sex at this developmental stage helps reduce biological variability and allows for clearer analysis of early-life asthma mechanisms. However, your point is valid—sex differences in asthma incidence reverse in adulthood, with women experiencing higher rates and severity. This shift is clinically important and warrants further investigation. Including girls in future studies or in parallel cohorts could help identify sex-specific risk factors and disease trajectories. We agree this should be addressed in the introduction to clarify the rationale and acknowledge the broader implications.
- What kind of tools were used to explore potential miRNA?
Answer: We used miRge3.0 tool to explore potential miRNA.
- An interesting part is that there are two possible groups based not on the markers and traditional asthma phenotyping/endotyping. Also, later in the manuscript, it is mentioned that the SAO group fitted the profile of eosinophilic or allergic asthma, while the other group fitted what? Paucigranulocytic asthma? Please present all data associated with the study subjects in the table, as it is unclear regarding CBC results, IgE levels, age, and the treatment they received.
Answer: We thank the reviewer and as suggested, we modified Table 1.
- Lines 59-60 contain the same information.
Answer: We corrected the statement as the following: The most common asthma variant in children, T2-high asthma, is characterized by atopy, eosinophilic inflammation, elevated Th2 cytokines (IL-4, IL-5, IL-13), and leukocyte recruitment driven by CD4+ T-cells, mast cells, and eosinophils.
- Interesting presentation of the results - in some cases, the data is presented in the middle of the sentence, even though there are no statistically significant differences. In other cases, it is mentioned that there are no differences, and the discussion ends. It would be ideal to review the results one more time and format them uniformly.
Answer: We thank the reviewer and edited the discussion as suggested.
Children with SAO were significantly older than those with NLF (p = 0.006). SAO patients showed higher FVC (p = 0.0197), indicating larger lung volume; however, their FEV1% predicted (p = 0.00028) and FEV1/FVC ratio (p = 1.42×10⁻⁷) were markedly lower, reflecting more pronounced airflow limitation. FEF75% predicted was dramatically reduced in SAO compared to NLF (p = 7.84×10⁻¹⁰), underscoring substantial small airway dysfunction. Additionally, IgE levels were significantly higher in the SAO group (p = 0.01), suggesting a stronger allergic component. No significant differences were noted in FVC% predicted (p = 0.579) or absolute FEV1 values (p = 0.427), indicating that while overall lung size may be similar, airway obstruction specifically distinguishes the SAO group.
- In lines 119-121, it is mentioned that the results fit the previous publications - can you highlight the novelty of the study in a separate segment? Additionally, this sentence would fit perfectly in the discussion section.
Answer: The statement indicates that the size and shape of the exosomes fit with standard exosomes characteristics that were previously published. However, there is no place to add the statement to the discussion. We appreciate the reviewer’s suggestion.
- P-values are often confusing and not presented evenly. Perhaps the exact p-value is not necessary to mention -20 degrees or other values, as it is sufficient to note that p < 0.0001.
Answer: We changed the p-values as suggested.
- The highlights in the tables where the significant differences are found could be in the tables- just a suggestion.
- Discussion and results presented well - no comments
- the 5.2 and 5.3 parts should be edited as there is no consistency - "lab", sometimes min, sometimes minutes, it is enough to mention "manufacturer's protocol used" if it is standard, no need to mention and cite different publications; sometimes x, "times" is used; also, please revise the minutes used for centrifugation and use previously mentioned abbreviations - TEM in 463-464.
Answer: We modified as it was suggested.
Round 2
Reviewer 2 Report
Comments and Suggestions for Authors
Thank you for accrediting the changes in the manuscript.